# Post-Vaccination Neutralization Responses to Omicron Sub-Variants

**DOI:** 10.3390/vaccines10101757

**Published:** 2022-10-20

**Authors:** Henning Jacobsen, Maeva Katzmarzyk, Melissa M. Higdon, Viviana Cobos Jiménez, Ioannis Sitaras, Naor Bar-Zeev, Maria Deloria Knoll

**Affiliations:** 1Department of Viral Immunology, Helmholtz Center for Infection Research, 38124 Braunschweig, Germany; 2International Vaccine Access Center, Department of International Health, Johns Hopkins Bloomberg School of Public Health, Baltimore, MD 21205, USA; 3Independent Consultant, Bogota 111111, Colombia; 4W. Harry Feinstone Department of Molecular Microbiology and Immunology, Johns Hopkins Bloomberg School of Public Health, Baltimore, MD 21205, USA

**Keywords:** SARS-CoV-2, Omicron, sub-variant, neutralization, COVID-19 vaccine

## Abstract

Background: The emergence of the Omicron variant (B.1.1.529), which correlated with dramatic losses in cross-neutralization capacity of post-vaccination sera, raised concerns about the effectiveness of COVID-19 vaccines against infection and disease. Several clinically relevant sub-variants subsequently emerged rapidly. Methods: We evaluated published and pre-print studies reporting sub-variant specific reductions in cross-neutralization compared to the prototype strain of SARS-CoV-2 and between sub-variants. Median fold-reduction across studies was calculated by sub-variant and vaccine platform. Results: Among 178 studies with post-vaccination data, after primary vaccination the sub-variant specific fold-reduction in neutralization capacity compared to the prototype antigen varied widely, from median 4.2-fold for BA.3 to 40.1-fold for BA.2.75; in boosted participants fold-reduction was similar for most sub-variants (5.3-fold to 7.0-fold); however, a more pronounced fold-change was observed for sub-variants related to BA.4 and BA.5 (10.4-fold to 14.2-fold). Relative to BA.1, the other Omicron sub-variants had similar neutralization capacity post-primary vaccination (range median 0.8-fold to 1.1-fold) and post-booster (0.9-fold to 1.4-fold) except for BA.4/5-related sub-variants which was higher (2.1-fold to 2.7-fold). Omicron sub-variant-specific responder rates were low post-primary vaccination (range median 28.0% to 65.9%) compared to the prototype (median 100%) but improved post-booster (range median 73.3% to 100%). Conclusions: Fold-reductions in neutralization titers were comparable post-booster except for sub-variants related to BA.4 and BA.5, which had higher fold-reduction. Assessment after primary vaccination was not possible because of overall poor neutralization responses causing extreme heterogeneity. Considering large fold-decreases in neutralization titers relative to the parental strain for all Omicron sub-variants, vaccine effectiveness is very likely to be reduced against all Omicron sub-variants, and probably more so against variants related to BA.4 or BA.5.

## 1. Introduction

In this third year of the COVID-19 pandemic, vaccines remain the most important tool to prevent severe disease in participants infected with SARS-CoV-2. Rapid evolution of SARS-CoV-2 resulted in the emergence of novel viral variants starting approximately one year after the beginning of the pandemic [1]. Vaccines with WHO emergency use authorization continue to show strong protection against severe disease, but protection against asymptomatic infection and mild disease due to the Omicron variant which emerged in November 2021 has been lower than for other variants [2]. The numerous mutations in its immuno-dominant spike protein raised concerns regarding its potential for immune-escape and in vitro analyses revealed that antibody recognition and function against the Omicron spike was severely diminished in vaccinated and convalescent participants infected with non-Omicron variants of SARS-CoV-2 [3,4,5]. Concurrent with these findings was the observation that antibody-mediated immunity waned relatively quickly, and waning protection and resurgent transmission became an issue in many populations [6,7,8]. Thus, the emergence of Omicron further motivated the administration of booster doses around the world.

Although the current booster vaccines were designed with the prototype (Wuhan-type) antigen, booster vaccination restored vaccine effectiveness against severe disease and resulted in a significant increase in antibody titers against Omicron, as well as an improved but still low neutralization of the Omicron-spike [3,6,9]. However, booster regimens failed to efficiently limit transmission or provide durable protection against symptomatic disease [2,10]. Furthermore, new clinically relevant sub-variants of Omicron continue to emerge with new mutations in the spike protein [11,12,13]. Because of this continuing rapid evolution of the main viral antigen, predicting the effectiveness of current vaccines or of new vaccines developed to counter these developments remains challenging. Clinical studies provide the best evidence on vaccine effectiveness but require time before data can be collected. Neutralization assays measure the ability of immune sera to inhibit viral entry into cells directly and are the most commonly used correlate of vaccine effectiveness [14,15,16]. Such in vitro data that assess the potential threat of these novel sub-variants provide more rapid insights and are usually the first available data to predict immune escape of new variants. Indeed, neutralization studies of post-vaccination antibodies demonstrated a dramatic reduction in cross-neutralization capacity against Omicron BA.1 that far exceeded the loss observed for previous variants of concern Alpha, Beta, and Delta [17,18,19]. Furthermore, a high proportion of vaccinees who received only a primary vaccine regimen failed to mount detectable antibody responses against Omicron BA.1, especially those immunized with vector-based or inactivated vaccines [4,20]. Moreover, neutralization studies have been instrumental in demonstrating that booster doses can restore functional antibody responses against Omicron [5,21,22]. Here, we summarize the existing literature on neutralization of all Omicron sub-variants with available data (BA.1, BA.1.1, BA.2, BA.2.12.1, BA.2.75 BA.3, BA.4/5, BA.4.6, BA.4.7, and BA.5.9), and we comparatively assess the sub-variant specific reductions in cross-neutralization compared to the prototype antigen.

## 2. Methods

We searched published and pre-print databases (PubMed, bioRxiv, and medRxiv) from 26 November 2021 (when WHO classified Omicron as a variant of concern) to 19 September 2022, for studies providing data on post-vaccination antibody responses to any Omicron sub-variant. As this work was performed as part of a broader ongoing literature review, broad search terms (“omicron” OR “BA.1.1.529”) were used; post-vaccination neutralization studies were identified by screening abstracts, which then underwent full text review.

Studies meeting the following criteria were abstracted: published or pre-print studies with neutralization data on at least one omicron sub-variant, assessing a WHO-authorized vaccine, and including samples collected less than six months after the last vaccine dose. Data on all age groups were included. Studies using surrogate neutralization assays, cohorts of immune-compromised participants (such as immuno-modulatory treatment or cancer) and cohorts preselected to high- or low-responders were excluded. Cohorts with hybrid immunity (any SARS-CoV-2 infection pre or post immunization) were excluded unless the proportion of hybrid-immune participants was below 20%. Completely naïve cohorts are increasingly difficult to obtain, and we aimed to include as many relevant studies as possible while minimizing the impact of hybrid immunity. For cohorts with unspecified infection status, hybrid-immunity was assumed low, and the study was included. 

Abstracted data included the type of neutralization assay, reference strain, number of specimens tested, number of doses, vaccine product(s)/regimen, timing of sample collection relative to final vaccine dose, neutralization titer for the prototype strain and for each Omicron sub-variant tested (GMT, median NT50, or mean NT50), and percentage of samples with detectable neutralization titers (responders) for the prototype strain and for each sub-variant. If multiple time-points beyond 2 weeks after the last dose were assessed within a cohort, the time-point closest to 2-weeks was included. If cohorts were measured using both live virus neutralization assay and pseudo-virus neutralization assay, the assay with more information (titers, responder rates) was included; if similar, live virus neutralization data were included. For cohorts that received a third dose of mRNA-1273 (Moderna), only results for 50 µg doses (the recommended booster regimen) were included; if the dosage was not specified, a 50 µg dose was assumed. Notably, some studies reported results for different clinical cohorts which we refer to as “observations”, thus one study may have more than one observation included in analyses.

Fold-reduction in neutralization titers for each sub-variant relative to prototype strain titers and for each sub-variant relative to all other sub-variants within the same study cohort were calculated for all study cohorts with relevant data; medians and interquartile ranges (IQR) of within-study fold-reductions were calculated across studies (medians and IQRs of log-transformed fold-reductions were first calculated and results then back-transformed). The median proportion of responders across studies by strain was also calculated. Because sub-variants BA.4 and BA.5 have identical spike sequences and the spike protein is the immune-dominant antigen of SARS-CoV-2, neutralization studies usually did not distinguish between BA.4 and BA.5 so these were presented combined as BA.4/5.

Results were stratified by vaccine platform (mRNA, vector, inactivated, protein subunit, heterologous immunization with at least one mRNA vaccine dose, and heterologous immunization without any mRNA dose) and regimen (primary versus booster). Comparisons between each of the omicron sub-variants using within-study comparisons of paired GMT results were not stratified by vaccine platform due to relevant studies being too few. The Kruskal–Wallis test was used to test for differences in median fold-reductions between vaccine platforms.

## 3. Results

Of 7688 studies screened, 251 met initial screening criteria and underwent full-text review, and 178 met inclusion criteria for abstraction. Of 178 studies, 145 (81.5%) assessed fold reduction of an Omicron sub-variant relative to the prototype strain, 43 (24.2%) assessed fold reduction of other Omicron sub-variants to BA.1, and 144 (80.9%) provided information on percent response to at least one Omicron sub-variant. An overview of the study selection process is shown in Figure 1.

Most data were available for mRNA vaccines, followed by inactivated vaccines, which had fewer observations, but a large number of sera analyzed due to several large cohorts. Regarding the sampling time-point of sera, 302 (73.4%) observations included were from sera acquired ≤1 month post vaccination, 75 (18.3%) observations were from sera acquired 1–3 months post vaccination and 34 (8.3%) observations were from sera collected >3–6 months post vaccination. Details on studies are provided in Appendix A.

### 3.1. Percentage of Samples with Titers above the Limit of Detection (Percentage of Responders)

First, we examined the percentage of samples with neutralization titers above the limit of detection (responders). Samples below the limit of detection are usually assigned a value between 0 and whichever limit of detection the used assay has. However, this can differ between studies and can affect the titer metrics on which fold-changes are based [23]. Among 373 observations from 135 studies with data after primary vaccination, responder rates ranged widely from 0% to 100% across all observations (Appendix A). The median percentage of responders differed by strain; there were fewer responders to Omicron sub-variants (medians ranged from 28.0% to 65.9%) than to the prototype strain (median 100%). The percentage of responders also varied by vaccine platform with fewer responders to Omicron sub-variants for vector-based and inactivated vaccines than for mRNA platforms (e.g., BA.1 strain: medians of 9.0% and 20.0% vs. 45.0%, respectively); the percentage of responders to the prototype strain did not vary by platform (median 100% across all platforms except for protein-based vaccines with 96.0%).

Responder rates to Omicron sub-variants greatly improved after booster vaccination, with medians above 100% for all sub-variants when analyzed irrespective of vaccine platform, except for BA.4.6 and BA.4.7 with 73.3% and 93.3%, respectively. However, only one study was available for each of these sub-variants. mRNA vaccines still tended to have more responders (e.g., 100% for BA.1 vs. 22.0% and 77.4% for vector-based and inactivated vaccines, respectively; Appendix A). However, wide heterogeneity between studies persisted (range 0 to 100%) and differences were not statistically significant.

### 3.2. Comparison of Omicron Sub-Variants to the Prototype (WT) Strain

Of the 178 included studies, 145 (81.5%) reported data on the fold-reduction of at least one Omicron sub-variant relative to the prototype strain. For primary series vaccination, the number of observations ranged from 129 for BA.1 to 2 for BA.2.75; no studies were available for BA.4.6, BA.4.7, and BA.5.9. For booster vaccination, the number of observations ranged from 154 for BA.1 to 2 for BA.4.7 (Table 1).

Of the 89 studies (202 observations) for primary vaccination, the fold-reduction relative to the prototype strain in neutralization titers was large for all Omicron sub-variants, ranging from median 4.2 (IQR: 3.3–17.7) for BA.3 to 40.1 (IQR: 28.3–56.7) for BA.2.75 and was observed across all available vaccine platforms (Figure 2A). Fold-reductions for BA.1 relative to the prototype strain appeared larger for mRNA vaccines (median 21.4-fold, IQR: 15.4–36.3) compared to vector (11.8-fold, IQR: 2.3–20.9) and inactivated vaccines (10.2-fold, IQR: 4.7–14.2; *p* ≤ 0.001), but mRNA vaccines had higher titers to the reference prototype strain (median 624.0 vs. 69.0 and 65.7, respectively; *p* < 0.001; data not shown); fold-reduction for heterologous regimens involving an mRNA platform was also large (21.8 fold) but was based on only 2 observations; no studies evaluated fold-reduction of BA.1 relative to prototype for other heterologous regimens (Appendix A).

Among the 116 studies (337 observations) for booster vaccination, fold-reductions in neutralization titers for Omicron sub-variants relative to the prototype strain were not as large as they were following primary vaccination: median reductions ranged from 5.3-fold for BA.2.75 (IQR: 4.8–7.2) to 14.2-fold (IQR: 11.9–14.2) for BA.4.6 (Figure 2C). Although median fold-reductions of BA.1 tended to be slightly larger for vector-based, inactivated, and protein-based vaccines (8.2-fold, 9.4-fold, and 10.0-fold, respectively) than for mRNA vaccines (6.0-fold), there was wide heterogeneity within strata, and IQRs overlapped broadly (Appendix A).

### 3.3. Comparison of Omicron Sub-Variants to Each Other

To better understand if immune escape is increased for some Omicron sub-variants more than others, fold-changes between sub-variants were compared in studies evaluating a booster dose; results for primary regimens were not evaluated due to the potential bias in fold-reductions resulting from high non-response rates. An overview of included studies and available observations is shown in Table 2.

Studies that assessed more than one sub-variant reported disproportionately lower fold-reductions in neutralization capacity for BA.1 relative to the prototype strain (Figure 2B,D). Therefore, we restricted our sub-variant specific analysis to studies providing matched observations for at least two-subvariants.

42 studies reported results on post-booster fold-changes for at least one pair of sub-variants, all of which evaluated BA.1 relative to another sub-variant; 35 studies evaluated three doses of mRNA vaccine, ten evaluated three doses of inactivated vaccine, five evaluated 3 doses of vector vaccines (or 2 doses if primary vaccine was Janssen-Ad26.COV2.S), two evaluated three doses of protein subunit vaccines, eight evaluated heterologous platform booster regimens which included an mRNA vaccine, six evaluated three doses of heterologous platform booster regimens without an mRNA vaccine, and two evaluated unspecified vaccine regimens (Appendix A). Some studies report data for different vaccine platforms, regimen or sub-variants and are therefore listed multiple times.

No notable differences were observed in neutralization titers between Omicron sub-variants when compared to Omicron BA.1, except for BA.4/5: median fold-reductions ranged from 0.9-fold (IQR: 0.7–1.2) for BA.1/BA.2 to 1.4-fold (IQR: 1.4–1.5) for BA.1/BA.2.75, whereas BA.4/5 had a median 2.1-fold (IQR: 1.5, 3.1) higher reduction relative to BA.1 (Figure 3A). Only one study was available providing data for BA.4.6 with a 2.7-fold reduction relative to BA.1 (IQR: 2.7, 2.7) and no studies were available for BA.4.7 and BA.5.9. When all combinations of non-BA.4/5 sub-variants were compared to BA.4/5, only minimal differences were observed across all sub-variants with available data ranging from 1.7-fold lower relative to BA.2.12.1 (IQR: 1.3–2.1) to 2.3-fold relative to both BA.1.1 (IQR: 1.7–2.9) and BA.2 (IQR: 1.8–2.9) (Figure 3B). No noticeable differences were observed in relative responses by vaccine platform (Figure 3A).

## 4. Discussion

The Omicron variant, with more than 30 mutations in the immune-dominant spike protein, spread rapidly and replaced the previously prevalent Delta variant in most regions of the world within weeks [24]. Viral evolution continued and only half a year after the emergence of Omicron, several clinically relevant sub-variants emerged and replaced the initial variant BA.1 [12,13,25]. The sub-variants BA.4 and BA.5 were especially successful in replacing BA.1 and in causing re-infections, even in Omicron convalescent participants [26]. Hence, the question arose of whether subsequent Omicron sub-variants showed increased immune escape compared to BA.1 or if these variants have other selection advantages over BA.1.

We reviewed all available literature reporting post-vaccination neutralization data for Omicron sub-variants up to 19 September 2022. The data landscape was highly uneven with far more evidence available for BA.1, the sub-variant that emerged first, and for mRNA vaccines. Fold-reductions relative to the prototype strain post-primary vaccination were large for all Omicron sub-variants, generally greater than 10-fold; neutralization capacity improved after the booster dose but was still lower against Omicron sub-variants compared to the prototype strain; generally 6- to 7-fold reductions were observed. The exception were variants related to BA.4/5 which had larger fold-reductions, approximately 23-fold post-primary and 12-fold post-booster relative to the prototype strain and 2-fold post-booster relative to BA.1. No other noticeable differences were observed among the other Omicron sub-variants. Notably, only very few studies were available for BA.2.7.5, BA.3, BA.4.6, BA.4.7, and BA.5.9 and summarizing results for these variants should be considered with care. We therefore focused our analysis on BA.4/5, for which robust evidence was available, but decided to keep data on newer subvariants to provide a first overview of their neutralization tendency.

Reductions were observed for all four types of vaccine platforms in use (and heterologous regimens); however, because neutralization responses and clinical effectiveness vary by vaccine platform, the overall results may not generalize to all vaccine types. mRNA vaccines generally showed on average the lowest fold-reductions (e.g., median fold-reduction post-booster for BA.1 was 6.0) while protein vaccines showed the highest (median for BA.1 was 10.0). There were few observations on boost regimens for vector-based (*n* = 5 for BA.1) and protein subunit based (*n* = 4 for BA.1) vaccine platforms. Results between studies evaluating the same vaccine platform/sub-variant often varied widely (e.g., the IQR for mRNA fold-reduction post-booster for BA.1 relative to prototype was 4.0-fold to 10.3-fold). Evidence for inactivated vaccines suggested larger fold-reductions (median fold-reduction post-booster for BA.1 was 9.4) than for mRNA vaccines (6.0 fold-reduction); although there were fewer observations for inactivated vaccines (*n* = 24 BA.1 data points vs. *n* = 86 for mRNA), some of the cohorts evaluated were large with more than 1000 sera evaluated.

A high percentage of participants had titers against Omicron sub-variants below the limit of detection. High proportions of “non-responders” render titer metrics and respective fold-changes artificial and potentially misleading, either because non-responder titers are usually given an arbitrary value below the limit of detection, or are excluded altogether, either way affecting the overall fold reduction calculation. Additionally, because the magnitude of a fold-change relative to the prototype strain is dependent on the prototype strain titers, if prototype titers are low after primary vaccination, there will be higher numbers of non-responders. As a consequence, fold-changes will be small and should not be compared to fold-changes based on initially high overall titers such as those usually obtained from post-boost donors, which may be larger despite higher neutralizing capacity. This was observed especially following primary vaccination with vector-based or inactivated vaccines, but not mRNA or heterologous regimens that included mRNA vaccines. Therefore, an important conclusion is that neutralization data against omicron-subvariants after primary series vaccination should be interpreted with caution. Consequently, we restricted our cross-sub-variant analysis to post-boost cohorts.

Another important finding was the wide heterogeneity in neutralization titers and responder rates reported across studies, both post-primary series and post-booster vaccination. These differences most likely arise from differences in methodology, such as the type of neutralization assay and their corresponding limits of detection, but also from differences in the cohorts. Our review included studies where sampling was performed up to six months after the last vaccination dose so effects of waning immunity over time may explain some of the heterogeneity observed, but not much since only 8.3% of observations evaluated antibody levels after 3 months since vaccination and we did not observe any obvious differences in these observations compared to data from earlier time-points (Appendix A). Differences between studies in the age distribution of participants and including studies with some hybrid immune participants albeit in low percentages may also account for some of the heterogeneity. Statistical variability also likely plays a role, since 20% of studies used had very small (*n* ≤ 10) sample sizes. To minimize study-to-study differences, we used observation-specific fold-changes in analyses which produce more robust estimates of variant-specific neutralization potency.

Overall, BA.1 results were different in studies that reported only on BA.1 vs. those that reported on both BA.1 and other sub-variants and were used to estimate relative fold-changes between sub-variants. Those studies that did not provide data on other sub-variants reported remarkably high fold-reductions for BA.1 while studies providing data on other sub-variants generally found smaller fold-reductions for BA.1. Our analyses comparing between sub-variants were restricted to only studies with direct comparisons to reduce between-study bias but may therefore not be fully representative.

Limitations of this study include lack of assessment of the effects of waning immunity, especially potential synergistic effects of waning immunity and increasing immune escape and their implications on vaccine performance. Most studies in this review evaluated immune responses primarily during peak antibody levels (i.e., within 2 months since vaccination) and conditions of waning immunity that have been observed after several months in clinical studies of vaccine effectiveness against Omicron infection may produce different results [6]. Moreover, data on hybrid-immune cohorts (those with SARS-CoV-2 infection(s) before or after immunization) were not included. It is well known that previous SARS-CoV-2 infections may strongly affect immunity and neutralization responses [17,27,28]. Additionally, it has been shown that effects on Omicron sub-variant immunity differ by whether the infection was caused by Omicron or another variant [26,29]. These complex effects must be considered to fully understand resulting implications for public health. Assessing participants with confirmed Omicron immune history (either by infection or Omicron-based vaccination) will be needed to fully understand cross-reactivity of Omicron sub-variant-directed neutralizing antibodies against antigenically different subvariants, including against new subvariants that are likely to arise. This will be especially important in the context of the new Omicron-containing bi-valent vaccines based on BA.1 and on BA.4/5 that are being deployed during a time of BA.5 prevalence.

Taken together, the overall reduction in neutralization responses compared to the prototype-strain was comparable (range 5.3-fold to 6.6-fold reduction) across all sub-variants except for sub-variants related to BA.4 or BA.5, which had increased fold-reductions. Our findings provide important implications for the prediction of immunity against Omicron-subvariants in the context of parental-spike directed immunity. We identify and discuss pitfalls when assessing neutralization data that will be crucial for appropriate evaluation and contextualization of novel vaccine immunogenicity data.

## Figures and Tables

**Figure 1 vaccines-10-01757-f001:**
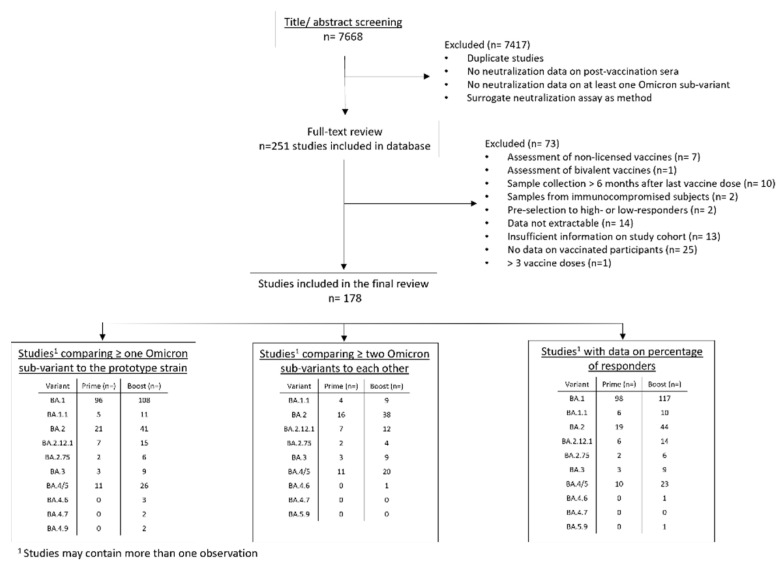
Overview of literature search and study selection process.

**Figure 2 vaccines-10-01757-f002:**
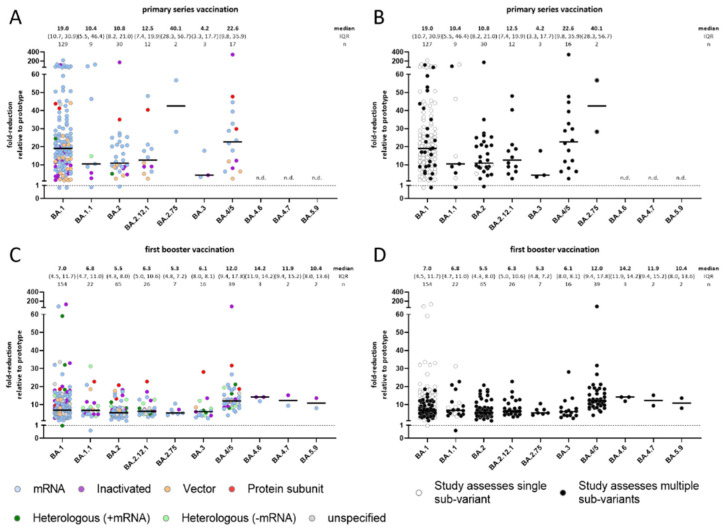
Fold-reduction in neutralization titers relative to the prototype strain for Omicron sub-variants. Primary (**A**,**B**) series vaccination or first boost vaccination (**C**,**D**), color coded by platform (**A**,**C**) or stratified for studies providing multiple variant-specific, paired observations (**B**,**D**). Every data point represents one observation. Median, IQR and group size is shown. n.d. no data.

**Figure 3 vaccines-10-01757-f003:**
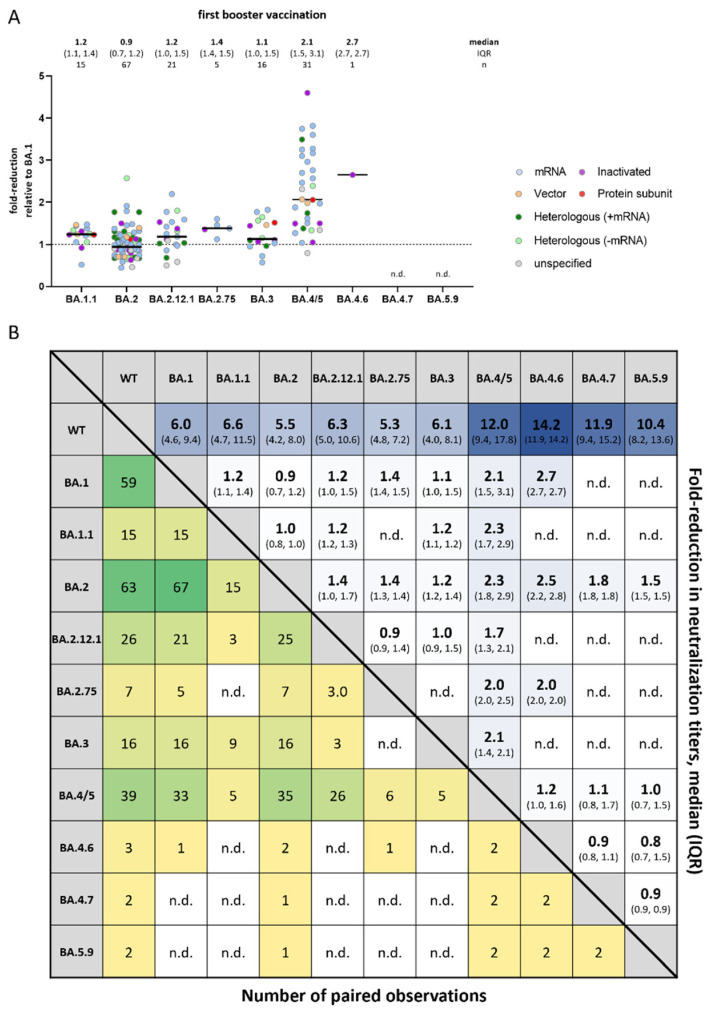
Relative fold-reduction in neutralization titers across Omicron sub-variants. Reported fold-change in neutralization titers of Omicron sub-variants compared to Omicron BA.1 (**A**) and for each sub-variant to each-other (**B**). Only studies with at least two sub-variant specific and matched observations are shown. Median and IQR (upper right corner) and number of paired observations (lower left corner) is shown. n.d. = no data.

**Table 1 vaccines-10-01757-t001:** Studies comparing Omicron sub-variants to prototype strain, availability of evidence per vaccine platform and Omicron sub-variant.

Vaccine Platform	Omicron Sub-variant Compared to Prototype	Primary Series Vaccination			First Booster Vaccination		
		Studies	Observations	Sera	Studies	Observations	Sera
**mRNA**	BA.1	69	87	1748	76	86	2368
	BA.1.1	3	6	103	6	8	119
	BA.2	17	20	323	33	34	629
	BA.2.12.1	5	6	124	12	12	255
	BA.2.75	2	2	35	5	6	134
	BA.3	2	2	20	6	6	97
	BA.4/5	8	9	164	20	21	738
	BA.4.6	0	0	0	1	1	15
	BA.4.7	0	0	0	1	1	15
	BA.5.9	0	0	0	1	1	15
**Inactivated**	BA.1	15	16	657	24	24	1118
	BA.1.1	2	2	33	5	5	156
	BA.2	4	4	59	9	10	229
	BA.2.12.1	2	2	37	4	4	101
	BA.2.75	0	0	0	1	1	40
	BA.3	1	1	10	3	3	76
	BA.4/5	3	3	49	6	6	153
	BA.4.6	0	0	0	2	2	80
	BA.4.7	0	0	0	1	1	40
	BA.5.9	0	0	0	1	1	40
**Vector**	BA.1	17	20	334	5	5	115
	BA.1.1	0	0	0	2	2	82
	BA.2	3	4	47	2	2	82
	BA.2.12.1	2	3	37	0	0	0
	BA.2.75	0	0	0	0	0	0
	BA.3	0	0	0	1	1	41
	BA.4/5	2	3	37	1	1	41
	BA.4.6	0	0	0	0	0	0
	BA.4.7	0	0	0	0	0	0
	BA.5.9	0	0	0	0	0	0
**Protein Subunit**	BA.1	2	2	39	4	4	96
	BA.1.1	0	0	0	1	1	20
	BA.2	1	1	10	1	2	40
	BA.2.12.1	1	1	10	1	1	20
	BA.2.75	0	0	0	0	0	0
	BA.3	0	0	0	1	1	20
	BA.4/5	2	2	39	2	2	68
	BA.4.6	0	0	0	0	0	0
	BA.4.7	0	0	0	0	0	0
	BA.5.9	0	0	0	0	0	0
**Heterologous (+mRNA)**	BA.1	2	3	114	14	18	1029
	BA.1.1	0	0	0	0	0	0
	BA.2	1	1	10	6	8	165
	BA.2.12.1	0	0	0	2	3	56
	BA.2.75	0	0	0	0	0	0
	BA.3	0	0	0	1	2	30
	BA.4/5	0	0	0	2	3	56
	BA.4.6	0	0	0	0	0	0
	BA.4.7	0	0	0	0	0	0
	BA.5.9	0	0	0	0	0	0
**Heterologous (-mRNA)**	BA.1	0	0	24	8	9	234
	BA.1.1	1	1	24	5	6	186
	BA.2	0	0	0	4	5	126
	BA.2.12.1	0	0	0	3	3	88
	BA.2.75	0	0	0	0	0	0
	BA.3	0	0	0	3	3	76
	BA.4/5	0	0	0	3	3	88
	BA.4.6	0	0	0	0	0	0
	BA.4.7	0	0	0	0	0	0
	BA.5.9	0	0	0	0	0	0
**Unspecified**	BA.1	1	1	48	6	8	321
	BA.1.1	0	0	0	0	0	0
	BA.2	0	0	0	2	4	79
	BA.2.12.1	0	0	0	1	3	35
	BA.2.75	0	0	0	0	0	0
	BA.3	0	0	0	0	0	0
	BA.4/5	0	0	0	1	3	35
	BA.4.6	0	0	0	0	0	0
	BA.4.7	0	0	0	0	0	0
	BA.5.9	0	0	0	0	0	0
**Total**	BA.1	106	129	2964	137	154	5281
	BA.1.1	6	9	160	19	22	563
	BA.2	26	30	449	57	65	1350
	BA.2.12.1	10	12	208	23	26	555
	BA.2.75	2	2	35	6	7	174
	BA.3	3	3	30	15	16	340
	BA.4/5	15	17	289	35	39	1179
	BA.4.6	0	0	0	3	3	95
	BA.4.7	0	0	0	2	2	55
	BA.5.9	0	0	0	2	2	55

**Table 2 vaccines-10-01757-t002:** Studies comparing newer Omicron sub-variants to BA.1 sub-variant, availability of evidence vaccine platform and Omicron sub-variant.

Vaccine Platform	Omicron Sub-variant Compared to BA.1	Primary Series Vaccination			First Booster Vaccination		
		Studies	Observations	Sera	Studies	Observations	Sera
**mRNA**	BA.1.1	3	4	55	6	6	100
	BA.2	16	18	282	28	29	520
	BA.2.12.1	5	6	124	10	10	195
	BA.2.75	2	2	35	3	4	74
	BA.3	2	2	20	6	6	97
	BA.4/5	7	8	164	17	18	751
	BA.4.6	0	0	0	0	0	0
	BA.4.7	0	0	0	0	0	0
	BA.5.9	0	0	0	0	0	0
**Inactivated**	BA.1.1	1	1	10	3	3	76
	BA.2	4	4	59	9	9	213
	BA.2.12.1	2	2	37	3	3	85
	BA.2.75	0	0	0	1	1	40
	BA.3	1	1	10	3	3	76
	BA.4/5	3	3	49	4	4	97
	BA.4.6	0	0	0	1	1	40
	BA.4.7	0	0	0	0	0	0
	BA.5.9	0	0	0	0	0	0
**Vector**	BA.1.1	0	0	0	2	2	82
	BA.2	2	3	37	4	4	161
	BA.2.12.1	2	3	37	0	0	0
	BA.2.75	0	0	0	0	0	0
	BA.3	0	0	0	1	1	41
	BA.4/5	2	3	37	2	2	49
	BA.4.6	0	0	0	0	0	0
	BA.4.7	0	0	0	0	0	0
	BA.5.9	0	0	0	0	0	0
**Protein Subunit**	BA.1.1	0	0	0	1	1	20
	BA.2	1	1	10	1	1	20
	BA.2.12.1	1	1	10	0	0	0
	BA.2.75	0	0	0	0	0	0
	BA.3	0	0	0	1	1	20
	BA.4/5	2	2	39	1	1	48
	BA.4.6	0	0	0	0	0	0
	BA.4.7	0	0	0	0	0	0
	BA.5.9	0	0	0	0	0	0
**Heterologous (+mRNA)**	BA.1.1	0	0	0	0	0	0
	BA.2	0	0	0	8	14	355
	BA.2.12.1	0	0	0	2	3	56
	BA.2.75	0	0	0	0	0	0
	BA.3	0	0	0	1	2	30
	BA.4/5	0	0	0	2	3	56
	BA.4.6	0	0	0	0	0	0
	BA.4.7	0	0	0	0	0	0
	BA.5.9	0	0	0	0	0	0
**Heterologous (-mRNA)**	BA.1.1	0	0	0	3	3	76
	BA.2	0	0	0	6	6	156
	BA.2.12.1	0	0	0	2	2	68
	BA.2.75	0	0	0	0	0	0
	BA.3	0	0	0	3	3	76
	BA.4/5	0	0	0	2	2	68
	BA.4.6	0	0	0	0	0	0
	BA.4.7	0	0	0	0	0	0
	BA.5.9	0	0	0	0	0	0
**Unspecified**	BA.1.1	0	0	0	0	0	0
	BA.2	0	0	0	2	4	79
	BA.2.12.1	0	0	0	1	3	35
	BA.2.75	0	0	0	0	0	0
	BA.3	0	0	0	0	0	0
	BA.4/5	0	0	0	1	3	35
	BA.4.6	0	0	0	0	0	0
	BA.4.7	0	0	0	0	0	0
	BA.5.9	0	0	0	0	0	0
**Total**	BA.1.1	4	5	65	15	15	354
	BA.2	23	26	388	58	67	1504
	BA.2.12.1	10	12	208	18	21	439
	BA.2.75	2	2	35	4	5	114
	BA.3	3	3	30	15	16	340
	BA.4/5	14	16	289	29	33	1104
	BA.4.6	0	0	0	1	1	40
	BA.4.6	0	0	0	0	0	0
	BA.5.9	0	0	0	0	0	0

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
