# Peer review of "Post-Vaccination Neutralization Responses to Omicron Sub-Variants"

_vaccines, 2022, doi:10.3390/vaccines10101757_

Round 1

Reviewer 1 Report

The manuscript Post-vaccination neutralization responses to Omicron sub-variants by Jacobsen H. et al. is an interesting, analytical, and updated review of the COVID-19 vaccine-induced protection against the Omicron variant of concern and sub-variants. This review is very important in the light of vaccine strategies to be adopted in the different world countries where Omicron variant is spreading, in particular the sub-variant BA.4/5. However, in order the message may be more clearly expressed, some points should be better explained.

In particular:

Pag. 6, line 1, “81 studies” and line 14, “97 studies”; actually, they seem to be 127 and 172, respectively, as it may be inferred by Table 1, total, the sum of the studies for the single sub-variants after primary series vaccination and first booster vaccination, respectively. Instead, the observations agree in the text and in the table, being 180 and 270, respectively.

Pag. 7, lines 9-16, “34 studies” are analytically split in: 25 studies pertaining to mRNA vaccines, 6 to inactivated vaccines, 5 to vector vaccines, 2 to protein sub-unit vaccines, 6 to heterologous vaccines with mRNA and 7 to heterologous vaccines without mRNA, lastly 2 to unspecified vaccines. The sum is 53 and not 34. Thus, if 34 should not be split in the 53 studies, the phrase should be more clearly rephrased to avoid any misunderstanding.

Pag. 7, line 17, “observation” should be plural as “observations”

Pag 7, line 23, “non-BA.4/5”; actually, it seems that BA.4/5 is compared with non-BA.4/5 other Omicron sub-variants, thus it should be modified accordingly, to prevent any misunderstanding.

Pag 9, last three lines. “Our review included studies where sampling was performed up to six months after the last vaccination dose so effects of waning immunity over time may explain some heterogeneity.” and Pag. 10, lines 13-15, “This review focused on immune responses primarily approximately 1 month after immunization and does not reflect the conditions of waning immunity that have been observed after several months in clinical studies of vaccine effectiveness against Omicron infection [6].” These two parts of the Discussion seem to be quite contradictory, thus I would eliminate the second phrase which does not add anything crucial. Out of the 153 reported studies, in 6 the post-vaccine time is not specified and, of the remaining 147, in 99 the post-vaccine time of observation may overcome 1 month, thus eliminating the second phrase seems to me more adherent to the analysis carried out by the Authors.

Supplementary Fig. 1 K and L are not cited and commented in the text; thus, they may be eliminated.

Supplementary Fig. 2 (the second) should be Supplementary Fig. 3.

Supplementary Table: it is very informative; however, the first column (Ref ID) is poorly clear. In fact, it is not clear which meaning have the numbers, considering that they do not refer to the position in the references, because the references are only 47, whereas the reported studies in the Table are 153. I think that if 153 are the analyzed studies they should appear in the references and these ID ref should correspond to the number of positions in the references. Alternatively, it is only misleading. The column of vaccines should be completed with a footnote legend for explaining the abbreviations.

Finally, in order the results of this analytical review may be immediately understandable, usable and applicable by the readers, I suggest that the Authors could express the results not only as fold-reductions of neutralization titers against Omicron sub-variants, but even as percent of residual putative antibody protection, so that the percent of reduced protection compared to the prototype antigen may be immediately appreciated.

Author Response

General note to all reviewers: We updated the literature review to September 19 and therefore provide updated data for the already reviewed results (numbers change only slightly, no meaningful changes observed but improved the overall robustness for most variants).  We now also include data for four additional sub-variants: BA.2.75, BA.4.6, BA.4.7, and BA.5.9. While very few studies are currently available for BA.4.6, BA.4.7, and BA.5.9 we believe that including them is important as they are of increasing interest to the field.

The manuscript Post-vaccination neutralization responses to Omicron sub-variants by Jacobsen H. et al. is an interesting, analytical, and updated review of the COVID-19 vaccine-induced protection against the Omicron variant of concern and sub-variants. This review is very important in the light of vaccine strategies to be adopted in the different world countries where Omicron variant is spreading, in particular the sub-variant BA.4/5. However, in order the message may be more clearly expressed, some points should be better explained.

  • Thank you for your positive and constructive feedback. Please find our answers to your comments below.

In particular:

Pag. 6, line 1, “81 studies” and line 14, “97 studies”; actually, they seem to be 127 and 172, respectively, as it may be inferred by Table 1, total, the sum of the studies for the single sub-variants after primary series vaccination and first booster vaccination, respectively. Instead, the observations agree in the text and in the table, being 180 and 270, respectively.

  • Please note that some studies provide data on multiple sub-variants and in few cases even multiple observations for the same sub-variant. The numbers in the main text refer to the number of individual studies while in the respective tables studies might be listed multiple times, if they provide data on different sub-variants (lines 117-118). Numbers were adjusted throughout the manuscript after updating the data sets for the extended literature review.

Pag. 7, lines 9-16, “34 studies” are analytically split in: 25 studies pertaining to mRNA vaccines, 6 to inactivated vaccines, 5 to vector vaccines, 2 to protein sub-unit vaccines, 6 to heterologous vaccines with mRNA and 7 to heterologous vaccines without mRNA, lastly 2 to unspecified vaccines. The sum is 53 and not 34. Thus, if 34 should not be split in the 53 studies, the phrase should be more clearly rephrased to avoid any misunderstanding.

  • Please see comment above. We have now added a sentence for clarification (line 236-237).

Pag. 7, line 17, “observation” should be plural as “observations”

  • Thank you, we have corrected this.

Pag 7, line 23, “non-BA.4/5”; actually, it seems that BA.4/5 is compared with non-BA.4/5 other Omicron sub-variants, thus it should be modified accordingly, to prevent any misunderstanding.

  • We have corrected the sentence (lines 244-245).

Pag 9, last three lines. “Our review included studies where sampling was performed up to six months after the last vaccination dose so effects of waning immunity over time may explain some heterogeneity.” and Pag. 10, lines 13-15, “This review focused on immune responses primarily approximately 1 month after immunization and does not reflect the conditions of waning immunity that have been observed after several months in clinical studies of vaccine effectiveness against Omicron infection [6].” These two parts of the Discussion seem to be quite contradictory, thus I would eliminate the second phrase which does not add anything crucial. Out of the 153 reported studies, in 6 the post-vaccine time is not specified and, of the remaining 147, in 99 the post-vaccine time of observation may overcome 1 month, thus eliminating the second phrase seems to me more adherent to the analysis carried out by the Authors.

  • Thank you for pointing this out. We reviewed the distribution of the time since vaccination across the studies. 4% of observations represented titers 1 months since vaccination and 91.75% were 3 months. Only 8.25% represented titers between 3-6 months after vaccination. The rest were mixed or unspecified. We also reviewed the titers of these observations with specimens between 3-6 months after vaccination and found no obvious differences compared to observations from 3 months. We revised the text to eliminate the contradiction as follows:
    • Added text to results section to describe the distribution of the data regarding the time since vaccination (lines 150-153)
    • Revised discussion to add text in bold: “Our review included studies where sampling was performed up to six months after the last vaccination dose so effects of waning immunity over time may explain some heterogeneity, but not much since only 8.3% of observations evaluated antibody levels after 3 months since vaccination.” (lines 325-328)
    • Revised 2nd phrase mentioned to: “Most studies in this review evaluated immune responses primarily during peak antibody levels (i.e., within 2 months since vaccination) and conditions of waning immunity that have been observed after several months in clinical studies of vaccine effectiveness against Omicron infection may produce different results [6].“ (Lines 344-347).

Supplementary Fig. 1 K and L are not cited and commented in the text; thus, they may be eliminated.

  • While we do not directly refer to these figures, they are part of generalized statements regarding the responder rates in boosted subjects (lines 164-169). We therefore decided to keep these figures for completeness.

Supplementary Fig. 2 (the second) should be Supplementary Fig. 3.

  • Thank you, we have corrected this.

Supplementary Table: it is very informative; however, the first column (Ref ID) is poorly clear. In fact, it is not clear which meaning have the numbers, considering that they do not refer to the position in the references, because the references are only 47, whereas the reported studies in the Table are 153. I think that if 153 are the analyzed studies they should appear in the references and these ID ref should correspond to the number of positions in the references. Alternatively, it is only misleading. The column of vaccines should be completed with a footnote legend for explaining the abbreviations.

  • Thank you, the “Ref ID” was indeed an unnecessary artifact for tracking studies in our data bases and is not necessary as the citation variables (author, journal and year) are sufficient to distinguish them for readers. We therefore removed this column.
  • We further added a legend explaining the three-letter vaccine code in a separate tab “READ ME”.

Finally, in order the results of this analytical review may be immediately understandable, usable and applicable by the readers, I suggest that the Authors could express the results not only as fold-reductions of neutralization titers against Omicron sub-variants, but even as percent of residual putative antibody protection, so that the percent of reduced protection compared to the prototype antigen may be immediately appreciated.

-               We agree that such a link to potential protection would be a valuable addition should one exist; however, no correlates of protection or “protective” neutralizing antibody titers have been accepted/established, so such a statement would be highly hypothetical and overstates what we know at this point. 

Reviewer 2 Report

This review is an exceptional detail focusing on live virus neutralising assay. The information is excellent and reasonable. However, this manuscript has not met the criteria for a systematic review that could be of higher quality than a traditional review.

I am impressed with the results that show all vaccination regardless of regimens and subgroups in each regimen. That would be great for comparison.

Comments.

1. Methods. This manuscript can add more information that laboratory assays were used "live" virus to make it more clear.

Author Response

General note to all reviewers: We updated the literature review to September 19 and therefore provide updated data for the already reviewed results (numbers change only slightly, no meaningful changes observed but improved the overall robustness for most variants).  We now also include data for four additional sub-variants: BA.2.75, BA.4.6, BA.4.7, and BA.5.9. While very few studies are currently available for BA.4.6, BA.4.7, and BA.5.9 we believe that including them is important as they are of increasing interest to the field.

  1. Methods. This manuscript can add more information that laboratory assays were used "live" virus to make it more clear.
  • Thank you. Please note that we provide information about the assay used for each study in Supplementary Table 1, column H (“Assay Type”). We added a new tab to the supplementary table “READ ME” explaining the abbreviations and if authentic or pseudo-typed viruses were used.

Reviewer 3 Report

 Overview and general recommendation:

In the research, the authors screen abstracts from published and pre-print databases from 11.26.2021 to 07.25.2022. Studies meeting their criteria are included in this review. They analyze the data of neutralization of all Omicron sub-variants (BA.1, BA.1.1, BA.2, BA.2.12.1, BA.3, and BA.4/5) based on vaccine platform and regimen. They find that in neutralization responses, the Omicron sub-variants show comparable reduction to prototype-strain, except that BA.4/5 show increased fold-reduction.

I find the paper is organized in a proper way and the results are well described. Major methods are well described in the manuscript and properly used in the research. The figures are well organized and presented in an appropriate way. However, I still think there is something to be improved. I suggest the authors to add some references and discussion to emphasize the significance and novelty of the research.

Major comments:

1.      As the authors mentioned in the manuscript, there are too few studies on some cohorts that I think it is hard to draw a conclusion.

2.      In discussion part. I suggest the authors to include more references and information to emphasize the importance and the prospect of this research.

3.      The authors summarize the current research on immune responses after vaccination. I think they should add more discussion about the significance and novelty. For example, why this research is more distinguished than other researches. And how this research will contribute to future disease diagnosis and vaccine development.

Minor comments:

1.      Page1 line31, it should be “…varied widely in post-primary vaccination but were comparable in post-booster….”.

Author Response

General note to all reviewers: We updated the literature review to September 19 and therefore provide updated data for the already reviewed results (numbers change only slightly, no meaningful changes observed but improved the overall robustness for most variants).  We now also include data for four additional sub-variants: BA.2.75, BA.4.6, BA.4.7, and BA.5.9. While very few studies are currently available for BA.4.6, BA.4.7, and BA.5.9 we believe that including them is important as they are of increasing interest to the field.

In the research, the authors screen abstracts from published and pre-print databases from 11.26.2021 to 07.25.2022. Studies meeting their criteria are included in this review. They analyze the data of neutralization of all Omicron sub-variants (BA.1, BA.1.1, BA.2, BA.2.12.1, BA.3, and BA.4/5) based on vaccine platform and regimen. They find that in neutralization responses, the Omicron sub-variants show comparable reduction to prototype-strain, except that BA.4/5 show increased fold-reduction.

I find the paper is organized in a proper way and the results are well described. Major methods are well described in the manuscript and properly used in the research. The figures are well organized and presented in an appropriate way. However, I still think there is something to be improved. I suggest the authors to add some references and discussion to emphasize the significance and novelty of the research.

Major comments:

  1. As the authors mentioned in the manuscript, there are too few studies on some cohorts that I think it is hard to draw a conclusion.
  • Thank you for your positive and constructive feedback. We agree that results should be interpreted with care if few studies are available. We have added on this in the beginning of the discussion (lines 282-287). To overcome this limitation for the most important sub-variants, we have decided to update the data presented by extending our time window for literature review to September 19 to include as many studies as possible and to further increase the relevance of our work.
  • We therefore provide updated data sets and also decided to include sub-variants BA.2.75, BA.4.6, BA.4.7, and BA.5.9. While very few studies are currently available for BA.4.6, BA.4.7, and BA.5.9 we believe that including them is important as they are of increasing interest to the field. However, results on these newer sub-variants should be interpreted with care and therefore we highlighted this limitation in the discussion.

  1. In discussion part. I suggest the authors to include more references and information to emphasize the importance and the prospect of this research.
  • It is important to acknowledge that there are currently no accepted correlates of protection available and therefore it is difficult to directly link our findings to clinical outcome (vaccine effectiveness). We believe that our findings are relevant for the development of new vaccines (lines 360-362) and for the rapid assessment and contextualization of newly emerging sub-variants (lines 67-72, 357-360, 366-367). We further discuss important limitations and pit falls when assessing neutralization titers which is important for the interpretation of such studies.

  1. The authors summarize the current research on immune responses after vaccination. I think they should add more discussion about the significance and novelty. For example, why this research is more distinguished than other researches. And how this research will contribute to future disease diagnosis and vaccine development.
  • We are providing a comprehensive and analytical overview (/review) of neutralization responses against novel and clinically relevant Omicron sub-variants. The main messages of our work are the reductions and differences among the different omicron sub-variants and the ultimative implications for vaccine development and assessment of newly emerging viral variants. We further identify and discuss low responder rates after primary vaccination as a strong contributor to inter-study heterogeneity making interpretation of variant-specific fold-changes in neutralization titers impossible. Lastly we emphasize to only perform variant-spedific comparisons using studies assessing multiple variants with the same clinical cohort to overcome inter-study related biases. We adjusted our abstract and discussion to emphasize these points more clearly.

Minor comments:

  1. Page1 line31, it should be “…varied widely in post-primary vaccination but were comparable in post-booster….”.
  • Please note that this sentence was removed during revision of the abstract (see comment above).

Round 2

Reviewer 3 Report

The authors include more studies in this review to improve the relevance of this review. They adjusted the abstract and discussion to further emphasize that their study will contribute to vaccine development and rapid assessment and contextualization of newly emerging sub-variants. I find that the authors have put considerable effort into addressing the reports of the referees. As a result, the paper is very much improved and I have no problem recommending it for publication.